# Melanins of *Inonotus Obliquus*: Bifidogenic and Antioxidant Properties

**DOI:** 10.3390/biom9060248

**Published:** 2019-06-24

**Authors:** Marina A. Burmasova, Aidana A. Utebaeva, Elena V. Sysoeva, Maria A. Sysoeva

**Affiliations:** Kazan National Research Technological University, 420015 Kazan, Russian Federation, Russia; aidana.utebaeva@gmail.com (A.A.U.); inonotus@yandex.ru (E.V.S.); oxygen1130@mail.ru (M.A.S.)

**Keywords:** melanins, *Inonotus obliquus*, bifidogenic, bifidobacteria, antioxidant activity

## Abstract

Extracts and melanins from *Inonotus obliquus* are widely used in medicine due to their high antioxidant properties. This study is dedicated to define the influence of the physicochemical and antioxidant properties of *Inonotus obliquus* melanins and their bifidogenic effects on *Bifidobacterium bifidum* 1 and *Bifidobacterium animalis subsp. lactis.* For this purpose, melanins precipitated from *Inonotus obliquus* aqueous extracts, obtained by a few methods, and separated melanin fractions by organic solvents were used. For the melanin physicochemical properties analysis spectrophotometry, electron paramagnetic resonance (EPR) spectroscopy and dynamic light scattering methods were applied. Melanins and their fractions difference in particle size and charge, antioxidant properties, and redox potential were revealed. It was shown that the redox potential, the size of melanin particles and the z-potential had maximum influence on bifidobacteria growth. The greatest activating effect on bifidobacteria was established by using melanin isolated from aqueous microwave extracts in concentrations of 10^−13^, 10^−10^, 10^−5^ g/cm^3^. The use of this melanin with antioxidant activity 0.67 ± 0.06 mg/g (expressed as ascorbic acid equivalent), and with redox potential −5.51 ± 2.22 mV as a prebiotic allowed the growth of *Bifidobacterium bifidum* 1 s to increase by 1.4 times in comparison with ascorbic acid by 24 h of cultivation.

## 1. Introduction

In modern pharmaceutical chemistry, much attention is attracted to the melanins from higher fungi and polypores in antioxidant applications [1]. The polypore, *Inonotus obliquus* (also known as chaga) is a widely used medicinal raw material. Extracts from this species are administered in gastro-intestinal tract diseases and complex anti-cancer treatments [2]. The therapeutic activity of the extracts is, to a great degree, provided by the main component–fungal melanin, the content of which is about 50–60% of extractive substances [3]. Fungus melanins consist of pigment parts in which terpenoids, steroids, neutral lipids, phenolic compounds and others, as well as proteins and polysaccharides are localized [4].

*I. obliquus* melanins form particles with a size of 400 nm or smaller. They possess paramagnetic properties (EPR signal with a g-factor equal to 2.0044, width of 5.7, and paramagnetic center concentration of 7.4 × 10^17^ spin/g [5]), and antioxidant activity up to 60 kC/100 g of melanin [6]. Previously in [7] it was shown that some fractions separated from *Inonotus obliquus* melanin stimulated the growth of *Bifidobacterium bifidum* 1. Bifidobacteria are widely used as probiotics for functional foods and medicines. With the aim to activate bacteria growth, the nutritional medium can be enriched with substances that perform the function of growth stimulators, such as cysteine, Tween-80, para-amino-benzoic acid and others [8,9]. While activated, these substances can either be added at the stage of nutrient medium preparation or directly into ready-made nutrient medium concurrently with inoculums. In fermented milk production, a preliminary stage of bifidobacteria activation (before adding to milk at the stage of its fermentation) is usually used, for example by keeping bifidobacteria in sterile milk at the temperature optimum for 3–4 h [10].

To intensify the bifidobacteria growth, antioxidants are often applied. In this case, the correct concentration selection of the substance that activates bacteria development and reproduction is necessary. For example, bifidobacteria and lactobacilli cells growth as well as biomass increasing by an order of magnitude over the control, was achieved by adding an amount 0.03–0.05 % of ascorbic acid into the nutrient medium [11].

Therefore, it is important to expand the usage possibilities of melanins as antioxidants for bifidobacteria growth activation and to identify their application features.

The aim of this study was to determine the interrelation between physical, chemical, and antioxidant properties of *I. obliquus* melanins and their bifidogenic effect to *Bifidobacterium bifidum* 1 and *Bifidobacterium animalis subsp. lactis* for the selection of melanin that can maximally improve the bifidobacteria activity at its concentration range of 10^−15^–10^−2^ g/cm^3^.

## 2. Materials and Methods 

### 2.1. Materials

Shredded chaga mushroom (*Inonotus obliquus*) were bought in the pharmacy chain (OOO “Fitofarm”, Anapa, Russia) and were extracted with water for further melanin obtaining. Freeze-dried strains of *Bifidobacteria B. bifidum* 1 and *B. animalis subsp. lactis* were applied in the study for the melanin bifidogenic effect assay. As the source of bifidobacteria medicinal products “Bifidumbacterin” (ZAO “Partner”, Moscow, Russia) and “Linex for children” (Lek d. d., Ljubljana, Slovenia) were used.

### 2.2. Melanin Production 

M1 and M2 melanins were obtained by sedimentation of aqueous extracts with hydrochloric acid [12] that were received by double maceration (M1) [13] and by using microwave treatment with further maceration (M2) [14]. M3 and M4 melanins were separated from the M1 melanin by butanol (M3) [15] and petroleum-ether (M4) [16] treatment.

### 2.3. Phenolic Content and Antioxidant Activity Determination 

Total phenolic content was determined by the photometric method with 4-aminoantipyrin [17]. The absorbance was measured at 500 nm using the spectrophotometer, “UNICO UV/VIS”. A calibration curve was prepared with thepyrocatechin standard solutions (0–30 µg/cm^3^).

Total antioxidant capacity was evaluated by the phosphomolybdenum method by plate reader INFINITE M200PRO (“TECAN”, Grödig, Austria). The standard curve of ascorbic acid was obtained for concentrations ranging from 0 to 0.1 mg/cm^3^ [18]. 

Antioxidant activity assessment was performed by “Color Jauza-01-AA” liquid chromatography (NPO Khimavtomatika, Moscow, Russia) with an amperometric detector. Quercetin was employed as a standard [19].

### 2.4. Inonotus Obliquus Melanin Assays 

Size and charge measurements of melanin particles were conducted at Zetasizer Nano ZS [20]. 

Electron paramagnetic resonance (EPR) analysis was performed at EPR-300 Bruker spectrometer (microwave power: 20, 5, 0.02 mW, frequency: 9.5 GHz, Ettlingen, Germany). The solid samples were studied at room temperature and atmospheric pressure. 

### 2.5. Microorganisms and Culture Media

For the *B. bifidum* 1 *and B.animalis subsp. lactis* cultivation the standard Blaurock medium was used [21].The incubation was carried out at a temperature of 37 ± 1 °C for 72 h by the method described in [21]. Quantitative analysis of bifidobacteria was conducted using the standard hydrolysate-milk medium, by the ten folds dilutions according to [21]. Specific growth rate was estimated by the growth curves pursuant to the technique [22].

*I. obliquus* melanins were added to the culture medium in concentration levels of 10^−15^ − 10^−2^ g/cm^3^ concurrently with the inoculums. Samples for analyses were taken in at 0, 6, 12, 24, 48 and 72 h of incubation.

### 2.6. Statistical Analysis 

Statistical calculations were performed using the software program “Statistica 6.0” (Tulsa, OK, USA). Values are expressed as means ± standard deviations from three independent experiments (*n* = 3) at 90% confidence level (*P* = 0.90)

## 3. Results

### Inonotus Obliquus Melanins Physicochemical Profile 

Physical and chemical properties of melanins used for the *B. bifidum* 1 and *B. animalis subsp. lactis* activation are represented in Table 1, Table 2 and Table 3.

Observed melanins formed colloidal systems in water, which were featured with dispersed phases, dominantly presented by large melanin nanoparticles with a size of mainly up to 400 nm (Table 1). The largest polydispersity index was indicated for the M1 melanin colloidal system. It’s dispersed phase was submitted by the particles of large, medium and small size. The M2 and M3 melanin colloidal systems appeared the closest by the dispersed phases particles size, but they significantly differed by the index of polydispersity and charge of the particles.

M3 melanin particles predominantly possessed negative charge, −31.5 mV. Although the other melanin particles prevailed with close values to negative charge, they also had positively charged particles, which affected both the average charge of the particles and the conductivity of the samples (Table 2). Colloidal systems with dispersed phases of M1 and M2 melanins had the closest properties on particles charges and conductivity.

The colloidal systems of M1 and M2 melanins had the closest antioxidant capacity (Table 3). In comparison with the M1 melanin, M3 melanin had more hydrophobic properties and M4 melanin had, to a greater degree more hydrophilic properties, that followed their colloidal systems total antioxidant capacity reduction. In addition, M1 and M2 melanins’ higher antioxidant capacity can be contributed to higher contents of the free phenolic substances in their colloidal systems.

A similar tendency was not observed among antioxidant ability of melanins, measured by amperometric detection. Probably, this type of activity is more depend on the particle conformation singularities in these melanins, which their EPR spectra analyses indirectly proved. Typical narrow signals of *I. obliquus* melanin EPR spectra, taken at 293 K are usually referred to as the organic compounds, for instance, phenol semiquinone-quinone [23], or oligomeric phenolic compounds that form stacks which are electrons traps [24,25]. M1, M2 and M3 melanins had a signal at g = 1.999, but they differed by intensity and line width (Table 4), and M4 melanin signal was displaced at g = 2.000. It is believed, that the second signal in the melanin spectra is due to complexes formed by organic compounds with an inorganic component (iron ions) [23]. This signal in the melanin spectra, obtained at this temperature (293 K) is sometimes absent, as in M2 melanin spectrum, or is asymmetric, as in M1 and M3 melanin spectra [23]. In M4 melanin spectrum, the second signal has a symmetrical line shape. This indicates the complex structural differences of organic compounds with metal between these melanins.

According to EPR spectra data, M1 melanin must have the highest antioxidant properties as its spectrum included two signals at g = 1.999 and g = 2.09 with the highest intensity among the observed melanins. That is corroborated by high values of antioxidant activity and total antioxidant capacity.

The M2 melanin demonstrated the lowest antioxidant activity (amperiometric detection), that can be explained by the presence of only one narrow signal in its EPR spectrum, with an intensity four and a half times lower than that of the M1 melanin. The fact that M2 melanin has the lowest oxidation reduction potential indicates that it is the strongest reducing agent, confirmed by its high total antioxidant capacity.

The lower M3 melanin antioxidant activity (two times lower compared to M1 melanin) is related to its lower paramagnetic properties. This is expressed by the fact that the narrow signal of M3 melanin EPR spectrum has lower intensity than that of M1 melanin (Table 4).

Antioxidant properties of the M4 melanin are several times lower than those of M1 melanin, which is corroborated by alterations of wide and narrow signals in the EPR spectrum of M4 melanin compared to other melanins (Table 3 and Table 4, Figure 1). M2 melanin has the highest total antioxidant capacity and is the strongest reducing agent (has lowest redox potential), so it is able to scavenge oxygen from the medium and, thus, can create right conditions for the growth of obligate anaerobic bifidobacteria.

As a control antioxidant for the bifidobacteria activation, ascorbic acid was chosen. The experiments with the same concentrations of ascorbic acid and melanins for the detection of their influence on bifidobacteria growth were conducted.

Bifidobacteria growth characteristics are shown in Table 5, Table 6, Table 7 and Table 8 and Figure 2.

According to Table 5 and Table 6 which illustrate the activation of the *B. bifidum* 1 with ascorbic acid and with *I. obliquus* melanins, it can be concluded that their use does not contribute to exponential growth rate increase of microorganisms, but by 24 h of cultivation, applying ascorbic acid and melanin M2 in a concentration of 10^−5^ g/cm^3^ allows activation of *B. bifidum* 1 growth. The amount of resulting viable cells augmented by 6.7% and 10.4% compared to the control. M2 melanin concentration reduction to 10^−13^ g/cm^3^ (Figure 2) by 24 h of cultivation allowed an increase in the viable cells number by 15.3% in comparison with the control. M3 melanin in concentration of 10^−10^ g/cm^3^ and M2 melanin concentration of 10^−5^ g/cm^3^ (Table 6) demonstrated close results, however, for further investigations, M2 melanin was chosen due to simpler production technology, with one stage, which allows energy to be saved [14]. Table 7 shows that the initial potential of the melanin-supplemented medium is more than what the control has, and by 24 h of incubation of *B. bifidum* 1 it becomes smaller compared to the control. That has a positive effect on the growth of microorganisms, perhaps because the medium redox potential changes in a narrower range.

The effect of M2 melanin on *B. animalis subsp. lactis* is more pronounced. Its application allowed the enhancing of the specific growth rate by two times (M2 melanin concentration 10^−5^ and 10^−2^ g/cm^3^), and the amount of viable cells by 9.7% and 11.0% compared to the control.

## 4. Discussion

*B. bifidum* 1 and *B. animalis subsp. lactis* are typically used probiotics in the pharmaceutical and food industries [26]. Melanin was used as a prebiotic for bifidobacteria activation because of its ability to serve a variety of functions. Thus, as it contains proteins and polysaccharides, melanin can be used as an additional source of carbon and nitrogen for the bifidobacteria. In addition, as an antioxidant, it can change the oxidation-reduction potential of the culture media and create conducive conditions for bacteria growth. Melanin can also associate with bifidobacteria, which helps their activity. M1 melanins are the part of *I. obliquus* aqueous extracts and are often used in various disease treatments [27]. M2 melanins are the components of aqueous extracts obtained by the developed technology, which allows a reduction in the process of their production by three times [14]. M3 and M4 melanins are produced from M1 melanin. Compared to initial M1 melanins M3 melanins are more hydrophobic, because part of hydrophilic compounds is removed from their composition, and M4 melanins are more hydrophilic, since a part of hydrophobic compounds is removed from them. Physical and chemical parameters of melanins are presented in Table 1, Table 2 and Table 3.

In accordance with the results obtained, high activating capacity of the M2 and M3 melanins in relation to *B. bifidum* 1 and *B. animalis subsp. lactis* may be due to the fact that they have close particle sizes with a charge about −30 mV, the oxidation-reduction potential of the medium is nearly −5 mV. The advantage of further usage of M2 melanin for bifidobacteria activation compared with M3 melanin is due to simpler producing technology.

Thus, the melanins high bifidogenic effect in comparison with ascorbic acid on the *B. bifidum* 1 growth may be associated with the culture medium’s large alteration, which was ensured by the oxygen negative effect decreasing their growth and activity by reducing the medium redox potential (Table 3 and Table 7 and redox potential of ascorbic acid 202.67 ± 0.95 mV). This occurs due to the melanin particles physicochemical properties, as well as to phenolic compounds being released from them.

It was established that *I. obliquus* melanins bifidogenic effect on bifidobacteria depends to a greater degree on the culture medium redox potential, the melanins’ particle size and z-potential.

The specific effect of M2 melanin on bifidobacteria was shown. M2 in concentrations of 10^−13^ g/cm^3^ and 10^−5^ g/cm^3^ promotes vital activity of the *B. bifidum* 1, increasing the viable cell amount by 24 h of cultivation compared to the control by 15.3% and 10.4%, in case of *B. animalis subsp. lactis* activation by 9.7% and 11.0% in concentrations of 10^−10^ g/cm^3^ and 10^−5^ g/cm^3^.

Higher bifidogenic effect of M2 melanin on the activity of *B. bifidum* 1 by 1.4 times more than that of ascorbic acid can be explained by the cultural medium redox potential change.

## Figures and Tables

**Figure 1 biomolecules-09-00248-f001:**
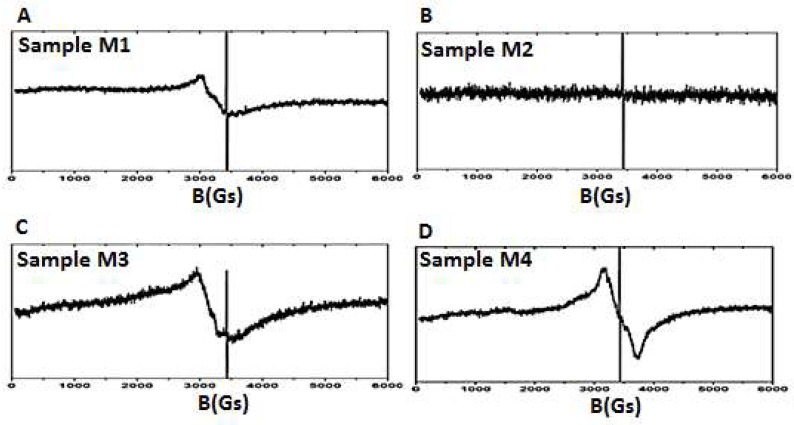
EPR spectra of M1 melanin (**A**), M2 melanin (**B**), M3 melanin (**C**), and M4 melanin (**D**).

**Figure 2 biomolecules-09-00248-f002:**
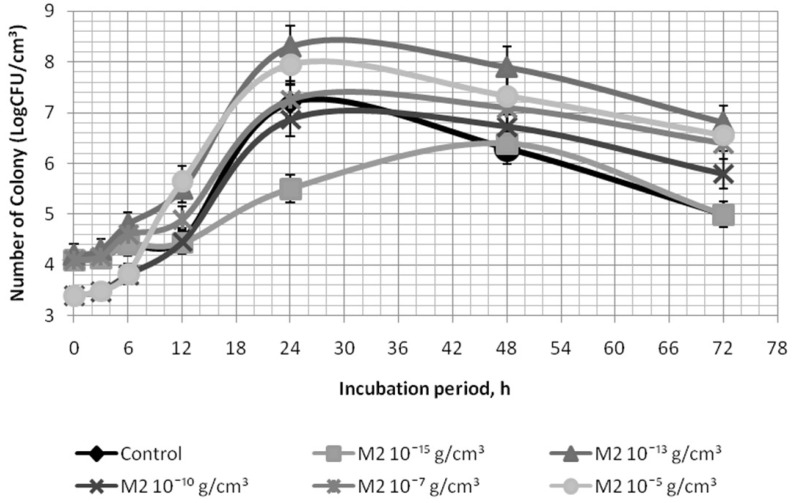
Growth curves of *B. bifidum* 1 in a medium supplemented with the M2 melanin.

**Table 1 biomolecules-09-00248-t001:** Polydispersity and particle sizes of *Inonotus obliquus* melanin aqueous colloid systems.

Object	Pd Index	Size and Quantity of Particles					
		Large		Medium		Small	
		Diameter, nm	Intensity, %	Diameter, nm	Intensity, %	Diameter, nm	Intensity, %
M1	1.00	356.0	71.8	64.65	16.0	22.87	10.3
M2	0.64	396.9 5360	80.4 1.4	42.64	18.2	-	-
M3	0.35	378.7 5371	81.7 1.2	96.88	17.1	-	-
M4	0.85	287.1	86.7	-	-	21.71 2.97	5.9 7.4

**Table 2 biomolecules-09-00248-t002:** Zeta potential and conductivity of *I. obliquus* melanin aqueous colloid systems.

Object	Conductivity, mS/cm	Zeta Potential						Zeta potential, mV
		High		Average		Low		
		Mean, mV	Area, %	Mean, mV	Area, %	Mean, mV	Area, %	
M1	0.27	138.0	0.6	18.2	9.5	−16.0	89.6	−14.00
M2	0.27	131.0	2.8	43.0	2.6	−27.6	87.1	−17.60
M3	0.71	-	-	-	-	−31.5	100.0	−31.50
M4	0.61	80.7	38.9	−12.4	7.9	−43.9	25.7	−1.34

**Table 3 biomolecules-09-00248-t003:** Antioxidative properties and phenolic content of *I. obliquus* melanin aqueous colloid solutions.

Melanin	Total Antioxidant Capacity, (Ascorbic Acid mg-Equivalent)/(g of Melanin)	Antioxidant Activity (Amperiometric Detection), mg/g	Oxidation-Reduction Potential of the Solution, mV	Total FREE Phenolic Substances, mg/cm^3^
M1	0.5555 ± 0.03	46.42 ± 2.39	4.07 ± 1.67	19.23 ± 0.75
M2	0.6650 ± 0.06	8.78 ± 0.67	−5.51 ± 2.22	22.20 ± 0.40
M3	0.2940 ± 0.01	21.90 ± 0.54	−4.16 ± 1.23	9.53 ± 0.30
M4	0.1113 ± 0.01	15.96 ± 2.89	323.50 ± 9.09	3.01 ± 0.14

**Table 4 biomolecules-09-00248-t004:** *I. obliquus* melanins EPR spectral characteristics.

Melanin	EPR Spectrum Signals				
	Wide		Narrow		
	g1	Int1	g2	w2 (Gs)	Int2
M1	2.09	3.33 × 10^7^	1.999	4.1	5.55 × 10^6^
M2	-	-	1.999	3.7	1.25 × 10^6^
M3	2.12	5 × 10^7^	1.999	5.0	3.15 × 10^5^
M4	1.99	4.62 × 10^7^	2.000	4.9	1.25 × 10^6^

**Table 5 biomolecules-09-00248-t005:** Growth measures of *B. bifidum* 1 activated by ascorbic acid, *n* = 3, *P* = 0.90.

Object	*B. bifidum* 1 Population Counts, lg CFU*/cm^3^				Exponential Growth Rate µ_,_ h^−1^
	Incubation Period, h				
	0	6	12	24	
Control	5.03 ± 0.04	5.20 ± 0.01	6.20 ± 0.02	8.37 ± 0.07	0.46
Ascorbic acid, 10^−10^ g/cm^3^	5.03 ± 0.03	5.27 ± 0.12	6.60 ± 0.13	8.30 ± 0.11	0.45
Ascorbic acid, 10^−5^ g/cm^3^	5.08 ± 0.11	5.28 ± 0.10	6.59 ± 0.06	8.93 ± 0.03	0.55

* colony-forming unit CFU.

**Table 6 biomolecules-09-00248-t006:** Growth measures of *B. bifidum* 1 activated by *I. obliquus* melanins, *n* = 3, *P* = 0.90.

Object	Concentration g/cm^3^	*B. bifidum* 1 Population Counts, lg CFU/cm^3^		
		Incubation Period, h		
		µ	0	24
Control		0.65	4.10 ± 0.27	7.20 ± 0.17
M1	10^−10^	0.73	4.46 ± 0.09	7.66 ± 0.10
M2		0.56	3.40 ± 0.16	6.87 ± 0.09
M3		0.69	3.66 ± 0.10	8.13 ± 0.10
M4		0.32	4.56 ± 0.09	7.06 ± 0.09
M1	10^−5^	0.35	4.92 ± 0.05	6.86 ± 0.10
M2		0.60	3.40 ± 0.16	7.95 ± 0.08
M3		0.54	4.53 ± 0.10	7.53 ± 0.10
M4		0.50	4.40 ± 0.16	7.66 ± 0.09

**Table 7 biomolecules-09-00248-t007:** The change in oxidation-reduction potential of the nutritional medium during the cultivation of *B. bifidum* 1.

Object	Concentration, g/cm^3^	Oxidation-Reduction Potential in the Medium, mV		Titer, lg CFU/cm^3^	
		0 h	24 h	0 h	24 h
Control	-	−125.28 ± 6.32	131.44 ± 6.66	5.02 ± 0.10	6.35 ± 0.21
M2	10^−5^	−16.02 ± 3.30	114.30 ± 4.63	5.24 ± 0.12	7.54 ± 0.20

**Table 8 biomolecules-09-00248-t008:** Growth measures of *B.animalis subsp. lactis* in nutritional medium supplemented with the M2 melanin, *n* = 3, *p* = 0.90.

Object	*B. animalis subsp. lactis* Population Counts, lg CFU/cm^3^				Exponential Growth Rate µ, h^−1^
	Incubation Period, h				
	0	6	12	24	
Control	6.07 ± 0.27	6.13 ± 0.05	7.95 ± 0.15	8.89 ± 0.04	0.70
M2, 10^−13^ g/cm^3^	6.07 ± 0.27	6.14 ± 0.39	7.35 ± 0.39	8.53 ± 0.25	0.29
M2, 10^−10^ g/cm^3^	6.07 ± 0.27	6.43 ± 0.39	7.70 ± 0.22	9.75 ± 0.14	0.51
M2, 10^−7^ g/cm^3^	6.07 ± 0.27	7.91 ± 0.08	8.53 ± 0.21	9.06 ± 0.40	1.49
M2, 10^−5^ g/cm^3^	6.07 ± 0.27	8.11 ± 0.04	9.05 ± 0.17	9.87 ± 0.09	1.61
M2, 10^−2^ g/cm^3^	6.07 ± 0.27	8.26 ± 0.01	9.05 ± 0.07	9.47 ± 0.26	1.86

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
