# Peer review of "Melanins of *Inonotus Obliquus*: Bifidogenic and Antioxidant Properties"

_biomolecules, 2019, doi:10.3390/biom9060248_

Round 1
Reviewer 1 Report
M.A. Burmasova and colleagues reported the physical properties of melanin in Inonotus obliquus and found anti-oxidant activity and growth support activity of bifidobacteria. This study may have merits. However, it is very difficult to understand for me (a biologist), WHAT (WHICH) melanin is helpful? No explanation was described before the specific terms, such as EPR-spectral, UFC/cm3, Zeta potential, Exponential growth rate µ, h-1. What we can judge from these analyses. Can the melanin remove hydroxy radicals? I recommend you perform biological experiments if you would like to conclude that the melanin has anti-oxidant activity. Are there any correlations between bacterial growth and anti-oxidant activity? Please add more explanations about your assay. I am sorry for not good at Twin-80 (Line37). Add references. . Remove a space in Line 119. Please rewrite your manuscript more kindly so that not only physicists, but also biologists, can understand your study.
Author Response
Thank you very much for objectively and close review of our article!
Point 1: WHAT (WHICH) melanin is helpful?
Response 1: Answering your question about which melanin is helpful, the corrections were made in article’s lines 30, 177.
Point 2: No explanation was described before the specific terms, such as EPR-spectral, UFC/cm3, Zeta potential, Exponential growth rate µ, h-1. What we can judge from these analyses.
Response 2:
We have made changes in text in the lines 79, 148 and tables 5, 6, 7, 8.accordingly to your comment: «No explanation was described before the specific terms, such as EPR-spectral, UFC/cm3, Zeta potential, Exponential growth rate µ, h-1. What we can judge from these analyses».
Zeta potential is the charge that develops at the interface between a solid surface – melanin and its liquid medium due to the presence of compounds with polar groups in melanin. This charge of melanin particles except cations and anions is directly connected with the electrical conductivity of the used colloidal systems. As was shown in the table 2, melanins M1 and M2 have similar values of Z-potential, electrical conductivity and total antioxidant activity (Table 3).
Exponential growth rate (µ) is a coefficient of proportionality between the cells increase rate and the present at that time bacteria number, characterizing growth rate of bacteria in the logarithmic growth phase. The higher µ shows more efficient of the microorganism’s growth.
The electron paramagnetic resonance spectroscopy (EPR) allows directly detect and quantify unpaired or odd electrons in atomic or molecular systems. So using EPR such paramagnetic centers as free radicals, that responsible for antioxidant activity of the object can be detected.
UFC/cm3 was mistake in the spelling and we have changed it to CFU/cm3. A colony-forming unit (CFU) is a unit used to estimate the number of viable bacteria in 1 cm3 of the sample.
Point 3: Can the melanin remove hydroxy radicals? I recommend you perform biological experiments if you would like to conclude that the melanin has anti-oxidant activity.
Response 3: The aim of the study was not antioxidant activity detection of chaga melanin "in vitro". We have been studied the activation of bifidobacteria using melanins as antioxidants.
Point 4: Are there any correlations between bacterial growth and anti-oxidant activity?
Response 4:
According to our experiments, the higher total antioxidant activity of the melanins provides higher growth rate.
The lower oxidation-reduction potential (redox potential) and oxidation-reduction properties of the object show the lower affinity for electrons, and thus a higher tendency to donate its electrons and be a reducing agent (neutralize a free radical). So, the most powerful antioxidants have low values of redox potential and oxidation-reduction properties. In our research the melanin with the lowest oxidation-reduction potential and oxidation-reduction properties has demonstrated the best growth rate.
Point 5: I am sorry for not good at Twin-80 (Line37). Add references.
Response 5: Sorry, it was mistake in the spelling and we have changed it to Tween-80 in the line 38.
Point 6: Please add more explanations about your assay. Please rewrite your manuscript more kindly so that not only physicists, but also biologists, can understand your study.
Response 6: According to your recommendations, we have made a number of edits throughout the text of the article.

Reviewer 2 Report
A paper on a potentially industrially relevant use of fungal melanin extracts would be a welcome addition to this special issue on melanins, but this manuscript is too hard to follow. A combination of minimal experimental detail provided and confusing text make it hard to tell if the conclusions are supported. This manuscript requires major revision before it can fully evaluated.
1. Editing for clarity in English is necessary. The content of one sentence often directs conflicts with the following sentence.
2. More information on the different melanin extracts is needed. This is published but not very accessible to non-Russian readers. Some information on the nature of these extracts in the Discussion but that should be earlier and in the manuscript. Chemical information, to the extent that is available, would be especially useful. For example, the single sentence that states that the fractions contain proteins and polysaccharides was unexpected.
3. The results for the growth experiments different fractions seem too similar to draw conclusions about which of the tested properties are important for promoting growth. It is unclear from the data presented why M2 was singled out for further experiments, and it is unclear which experiments are being referred to for the key comparison of M2 and ascorbic acid.
4. In terms of total anti-oxidant capacity, on a per gram basis each melanin fraction is less effective than ascorbic acid (values in Table 3 Column 1 are less than 1), so other factors are likely responsible for the greater growth promoting effects. The authors point to redox potential of the growth solution, but this is only reported for M2 and isn’t reported for ascorbic acid.
Author Response
Thank you very much for objectively and close review of our article!
Point 1: Editing for clarity in English is necessary. The content of one sentence often directs conflicts with the following sentence.
Response 1: The english spelling and style was checked by the specialist in this area.
Point 2: More information on the different melanin extracts is needed. This is published but not very accessible to non-Russian readers. Some information on the nature of these extracts in the Discussion but that should be earlier and in the manuscript. Chemical information, to the extent that is available, would be especially useful. For example, the single sentence that states that the fractions contain proteins and polysaccharides was unexpected.
Response 2: According to your recommendations, we have made a number of changes through the text of the article.
Point 3: The results for the growth experiments different fractions seem too similar to draw conclusions about which of the tested properties are important for promoting growth. It is unclear from the data presented why M2 was singled out for further experiments, and it is unclear which experiments are being referred to for the key comparison of M2 and ascorbic acid.
Response 3: According to your recommendations changes were made in the lines 143 and 161.
Point 4: In terms of total anti-oxidant capacity, on a per gram basis each melanin fraction is less effective than ascorbic acid (values in Table 3 Column 1 are less than 1), so other factors are likely responsible for the greater growth promoting effects. The authors point to redox potential of the growth solution, but this is only reported for M2 and isn’t reported for ascorbic acid.
Response 4: According to your recommendations, we have made a number of changes through the text of the article.
An experiment was not conducted with ascorbic acid, as redox potential of the ascorbic acid solution exceeds the potential of the melanin M2.

Reviewer 3 Report
This is an important study about melanins of inonotus obliquus. The description about Tables 5-8 can be improved to explain more in detail about P value in the Tables.
Author Response
Thank you very much for objectively and close review of our article!!
Point 1: This is an important study about melanins of inonotus obliquus. The description about Tables 5-8 can be improved to explain more in detail about P value in the Tables.
Response 1: According to your remark, corrections were made in the lines 93-94.

Round 2
Reviewer 1 Report
Now, the MS is improved.
Author Response
Point 1: I would not like to sign my review report
Response 1: Please, advise the reason for your refusal
Reviewer 2 Report
This manuscript is improved but there are still several confusing parts:
1. Where exactly are the comparisons for “Higher bifidogenic effect of M2 melanin on the activity of B. bifidum 1 and B. animalis subsp.lactis in 1,4-2,3 times more than of ascorbic acid” coming from? As far as I can tell, B. animals subsp.lactis wasn’t tested with ascorbic acid here.
2. I don’t understand this part of the author’s response to Point 4 from my initial review: “An experiment was not conducted with ascorbic acid, as redox potential of the ascorbic acid solution exceeds the potential of the melanin M2.” That would seem to indicate that (a) they have done the experiment, or (b) it is in the literature and could be referenced.
3. The sentence that begins on line 170 directly contradicts the next sentence (beginning on line 172).
4. The meaning of the sentence on lines 181-182 is unclear.
5. In line 192, “fixing” of nitrogen?
Author Response
Thank you for your diligence in our article reviewing. We agree with all points and have made necessary corrections
Point 1: Where exactly are the comparisons for “Higher bifidogenic effect of M2 melanin on the activity of B. bifidum 1 and B. animalis subsp.lactis in 1,4-2,3 times more than of ascorbic acid” coming from? As far as I can tell, B. animals subsp.lactis wasn’t tested with ascorbic acid here.
Response 1: The text was corrected in lines 21-23, 206-209, 217-219.
Point 2: I don’t understand this part of the author’s response to Point 4 from my initial review: “An experiment was not conducted with ascorbic acid, as redox potential of the ascorbic acid solution exceeds the potential of the melanin M2.” That would seem to indicate that (a) they have done the experiment, or (b) it is in the literature and could be referenced.
Response 2: We have done the experiment. Corrections were made in the lines 206-209
Point 3: The sentence that begins on line 170 directly contradicts the next sentence (beginning on line 172).
Response 3: The corrections were made in the lines 170-183
Point 4: The meaning of the sentence on lines 181-182 is unclear.
Response 4: The corrections were made in the lines 170-183